# Application of a 3D Scanner in Robotic Measurement of Aviation Components

**Krzysztof Kurc** [1,*] , **Andrzej Burghardt** [1] , **Piotr Gierlak** [1] , **Magdalena Muszyńska** [1] , **Dariusz Szybicki** [1] , **Artur Ornat** [2] and **Marek Uliasz** [2]

1. Department of Applied Mechanics and Robotics, Faculty of Mechanical Engineering and Aeronautics, Rzeszów University of Technology, al. Powstancow Warszawy 8, 35-959 Rzeszow, Poland
2. Pratt and Whitney Rzeszow S.A., 35-078 Rzeszow, Poland
* Correspondence: kkurc@prz.edu.pl; Tel.: +48-(17)-8651814

**Abstract:** The aviation industry is associated with high precision and accuracy standards of the manufactured components, and thus the need to ensure precise quality control. Measurement processes, depending on the manufactured components, take place before, during and after the processing stage. Optical scanners can be used for these measurements, the measurement results of which can be displayed on the operator panel or used to prepare a report. The innovative approach is to measure, compare the results with a pattern, send the deviations to a neural decision-making system, select the forces and send the results to a robot controller for adaptive machining. The presented proprietary solution includes a data acquisition system, a neural decision-making system and a robot that carries out the machining process via force control. The proposed solution was verified on aviation components. During the process parameter optimization stage for the diffuser and ADT gearbox, the points describing the change in width of the chamfer being performed and the blade thickness in the control sections were approximated.

**Keywords:** industrial robots; optical scanner; programming; neural decision-making system; communication

## 1. Introduction

The production of components for the aviation industry is associated with high precision and accuracy standards in the manufacture of components, and thus the need to ensure quality control at various stages of production. Measurement processes, depending on the manufactured semi-finished products, take place before, during and after the processing stages. For control and final measurements, various measurement techniques can be used, from taking a shape imprint and displaying the deviation value on a measuring projector, non-contact, e.g., with a laser profilometer or optical scanner, and by contact with the use of a probe and Coordinate Measuring Machine (CMM).

Taking the imprint and then cutting it and displaying it on the measurement projector is very accurate, but time-consuming, and the accuracy of the measurement depends on the correctness of manual cutting operations. The manual method with the laser profilometer requires the correct positioning of the measuring head in relation to the measured detail. Conducting measurements with coordinate machines requires additional time to mount the workpiece in the measuring area of the CMM. The advantages of this solution are high versatility and accuracy, while the disadvantages are the high cost of purchasing and maintaining the CMM and the long measurement time [1,2]. The aforementioned methods require a large amount of work time, while the detail must be inspected sometimes and in several hundred places, some of them are susceptible to the so-called human factor and do not allow automatic generation of a measurement report.

For the above reasons, it was decided to design and build a stand for conducting robotic measurements using a 3D optical scanner. Communication between devices and

data exchange represents a very important stage in the design and construction of robotic stations, and details the integration of machine systems and the introduction of changes in production processes aimed at increasing efficiency and introducing adaptation possibilities. Machining, e.g., grinding, are examples of processes that require common communication between many station components in a fast and continuous manner. While browsing the literature, one can find examples of studies on machining by robots [3,4]. Force control algorithms and their applications during processes have been presented in [5,6]. Communication between many robots, and surface shaping using an industrial robot and tool condition monitoring are described in works [7,8]. The authors of works [9–11] use the Profibus protocol for communication. The TCP/IP protocol is employed by the authors of [12–16]. Various programming languages can be used to create a dedicated network, with one increasingly popular example being Python [17,18], used by the authors in the application [19]. The authors of the article [20] describe the improvement in the grinding process of, for example, metal molds with pressure control using CAD/CAM data, and the improvement of the process by scanning is presented in articles [21–23]. Laser displacement sensor in the application of aero-engine blade measurement is presented by authors of the work [24]. The authors of the work [25] presented an innovative 3D measurement system for the material removed from the engine blades. The use of the GOM 3D scanner to measure the geometry of the blades in the process of inspection, servicing and repairs is presented in the works [26].

## 2. Measuring System and Robotic Station

The applied three-dimensional optical scanner obtains geometric data from an existing physical object. These data are used to reflect a virtual 3D model of the scanned object that can be used for a variety of applications such as reverse engineering, rapid prototyping, quality control, and cultural heritage documentation. Robotized geometry measurements are performed using the Atos Core 3D scanner (GOM: Oberkochen, Germany) and the Atos Professional software (GOM 7.04: Oberkochen, Germany). The 3D scanner is mounted on the ABB IRB 1600 manipulator, which allows for any positioning of the scanner head in front of the measuring element.

The Atos Professional software allows for recording the position and measurement orientation of the 3D scanner in space in order to automate and repeat the measurements of subsequent components. After completing part of the scan, the scanner moves and rotates to areas not recorded in the previous scan. The individual measurements are automatically processed using reference points relative to one common coordinate system, resulting in a complete 3D point cloud.

The Atos Core 3D scanner contains a fringe projector and two stereoscopic cameras, working on the triangulation principle (calculating the intersection of a plane and a ray in space). These fringes are recorded by two cameras shown in Figure 1, which creates a chamfer shift based on the sinusoidal distribution of the intensity on the camera matrices. Atos Core uses multiple chamfer shifts based on the heterodyne principle to achieve the highest sub-pixel accuracy. Based on the optical transformation equations, independent 3D coordinates are automatically calculated for each camera pixel.

The computed polygon mesh describes free-form surfaces and geometric elements. Its verification is possible by comparing the surface with the technical drawing or directly with the CAD data set. The software also enables the implementation of 3D surface analysis and 2D analysis of sections or points.

The Atos system measures the deviation between CAD data and actual 3D coordinates, providing full-field measurement data. These data, in addition to the representation of surface deviations in relation to CAD data, contain complete information about the object from which the software automatically determines other details. After the scanning stage and obtaining a 3D point cloud, it is possible to generate a report of the properties of a given scan as in Figure 2. The accuracy of the scanner depends on many factors, e.g., whether

one scan is sufficient for the measurement or several scans are needed, and if so, whether or not reference points were used to connect (stick) them.

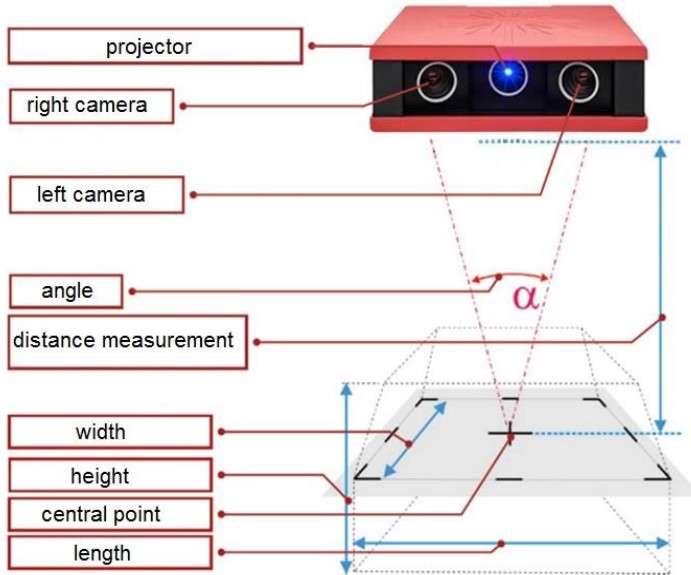

**Figure 1.** Atos Core—the idea of measurement.

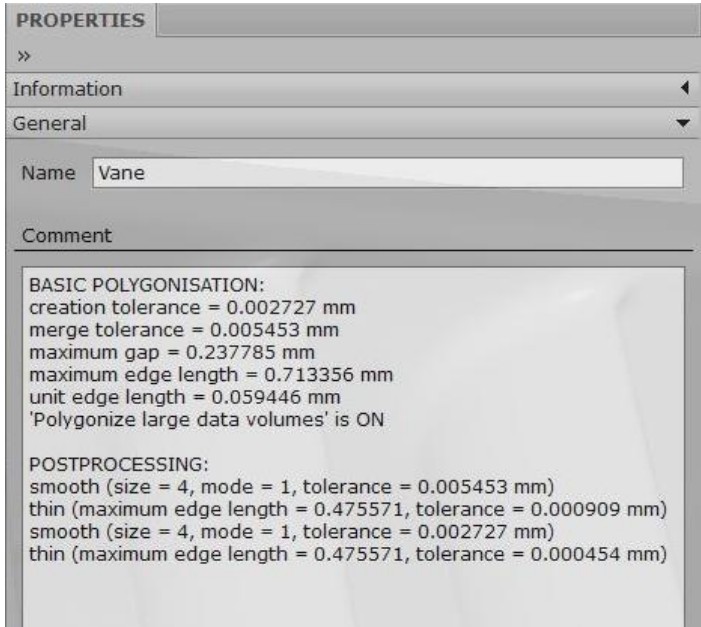

**Figure 2.** 3D-scanning properties.

Optical measurement systems owe their accuracy to the use of the latest solutions in the field of optoelectronics, precise image-processing technologies and mathematical algorithms, combined with unchanging precision standards and automatically performed calibration.

A novelty in the discussed issue is the robotic acquisition of a scan of the aerial part before processing, in comparison with the pattern, selection of the grinding pressure force using a neural decision-making system and sending the processed information in automatic mode to the robot controller in order to adapt the processing parameters.

The designed robotic station shown in Figure 3 for grinding aviation elements displays: the IRB 140 robot holding the detail, the electrospindle and the IRB 1600 robot with a 3D scanner. The Atos Professional software works with the Atos Core 3D scanner and communicates with the IRC5 robot controller via the TCP/IP protocol.

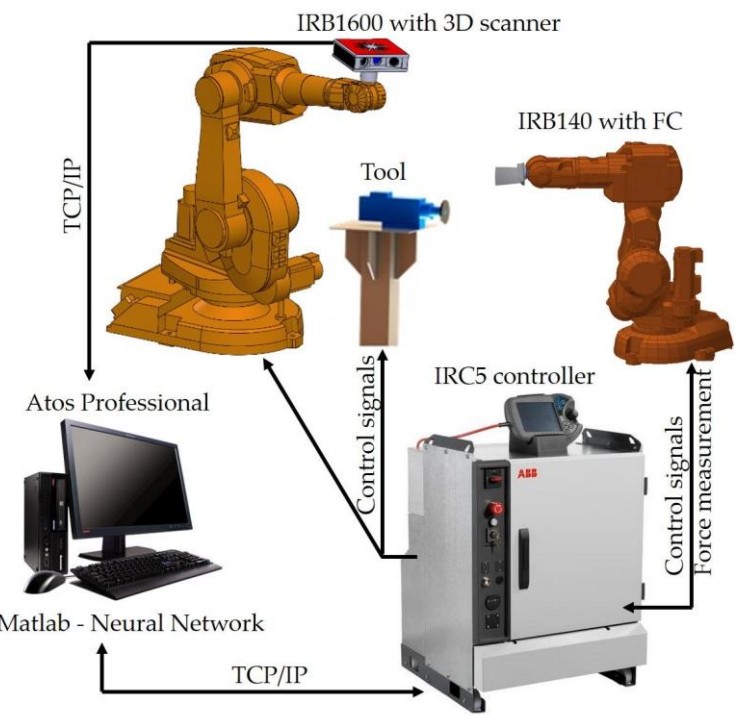

**Figure 3.** Schematic diagram of communication between robotic cell devices.

The RobotWare robot controller software (ABB 6.08: Zurych, Szwajcaria) has a force control option and provides control of two robots. The value of the force with which the detail is pressed against the electro-spindle tool determines the precise neural network based on information from the 3D scanner.

Atos Professional performs the tasks of the measuring system and, together with the Matlab software (2021B: Natick, MA, USA), communicates with the IRC5 robot controller by sending data via the TCP/IP protocol. The grinding tool speed is regulated by a frequency converter connected to the robot controller using DeviceNet.

The initial scenario for a robotic process is:

- Start;
- Measurement with a scanner;
- Comparison of the point cloud with the pattern;
- Data recording with deviations;
- Sending deviations to Matlab;
- Selection of the pressure force with a neural decision system;
- Sending received values to the controller;
- Automatic correction of the contact pressure for the programmed path;
- Machining;
- Control measurement by comparing with the reference (Yes/No);
- Stop and detail change.

## 3. Comparison of the Point Cloud with the Pattern

One of the components selected for testing is an aircraft blade. The blades are elements of gas turbine engines. Their shape makes it possible to convert the energy of the flowing medium into mechanical energy. The production of blades is carried out mainly for the aviation and energy industries. They are produced by the monocrystalline casting process, made on CNC machines, up to electrochemical drilling. Shaping the blades by electrochemical drilling is cost-effective, but the technology is sensitive to a number of factors, such as temperature, the intensity of the current flowing, the width of the gap, etc.

The uniqueness of the blade shape requires an individual approach for each piece during the grinding process.

For geometric measurement of the blades, contact and non-contact methods are used. The use of the contact method consists in the use of measuring probes to determine the geometric dimensions at selected points. The authors of [27,28] present this type of probe in various applications. There are also solutions with measuring systems placed on a robot [29]. The authors of the work [30] use coordinate measuring machines (CMM) to measure the grinding process. In non-contact measurements, laser heads [24] or optical scanners [15] are used. Optical scanners are mounted on stands [4,25] or on robot arms [31]. Contact devices are not sensitive to: dust, glare and lighting, but are slow compared to the number of measurements collected using non-contact methods.

Figure 4 shows a view of the blade after electrochemical drilling before grinding. Measurement with a 3D scanner as in Figure 5 allows one to measure the blade at each point of interest.

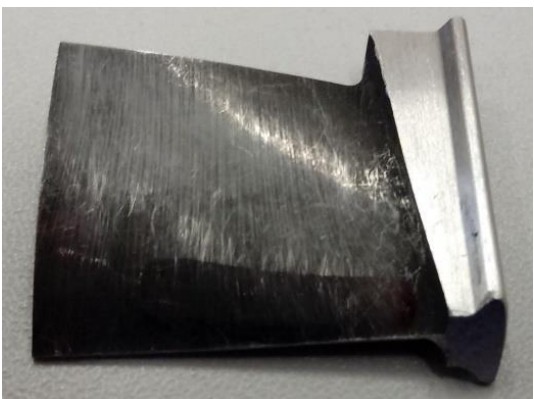

**Figure 4.** A blade before the griding operation.

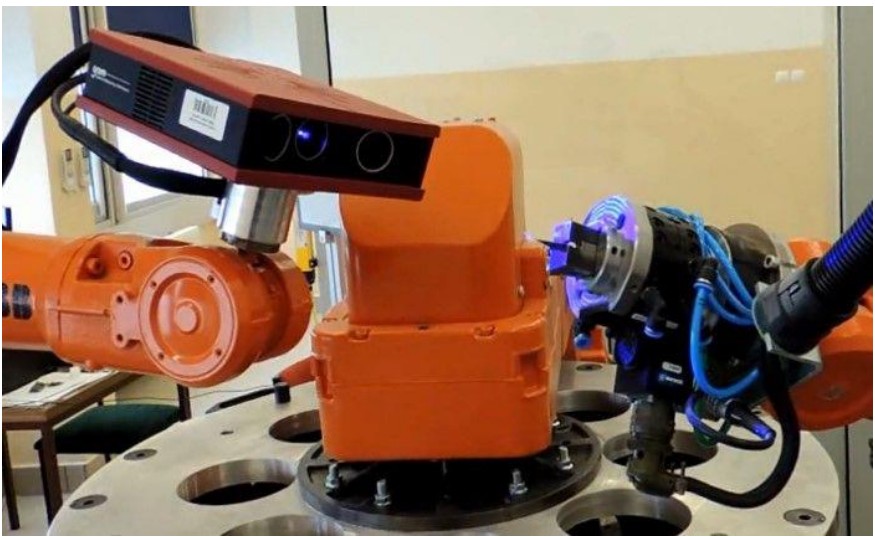

**Figure 5.** Measurement of blade geometry with the use of a 3D scanner.

The scanning program in the Atos Professional application (example shown in Figure 6) is an extensive hierarchical structure of dependencies between the nominal element (CAD pattern) and the current element (scan).

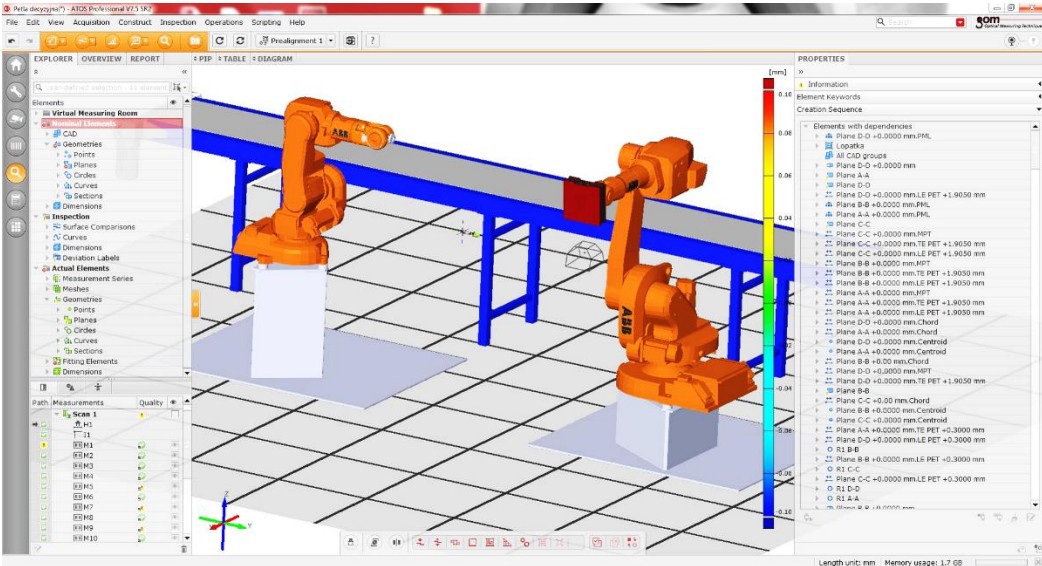

**Figure 6.** Virtual environment scanning program in the Atos Professional application.

In the Atos Professional software, after comparing the scan with the CAD model, allowances in 80 programmed points are determined shown in Figure 7.

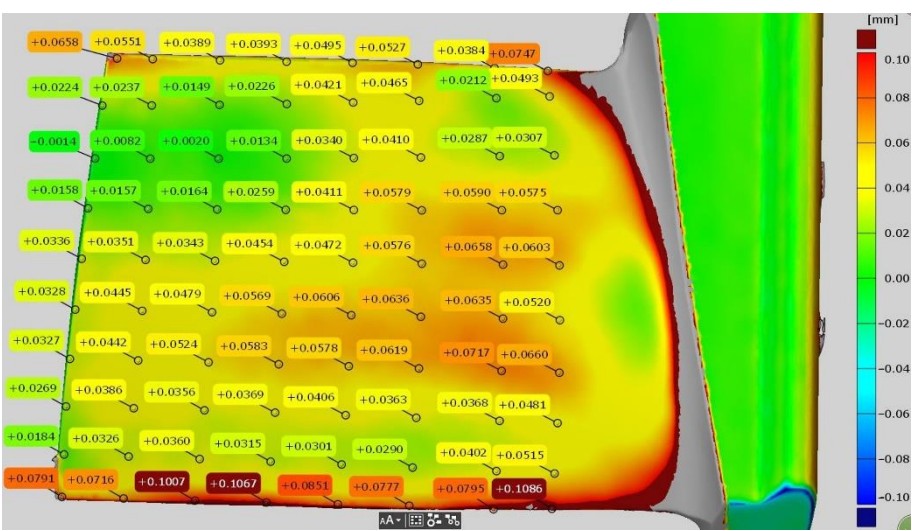

**Figure 7.** Determined allowances at 80 points on the blade.

Then the obtained dimensions are automatically saved in the variables defined in Python. In the process of determining the allowances, the Atos Professional Blade software add-on is used, dedicated to the measurement inspection of all types of blades.

## 4. Recording Data with Deviations and Sending It to Matlab

With the Atos Professional software, it is possible to save sequentially executed commands thus creating a macro. Recorded macros can be called automatically and they can contain advanced program functions shown in Figure 8. This allows a recorded sequence of operations to be performed repeatedly. By editing a recorded script, it can be adapted to other tasks or generalized. Script programming is based on modifying or combining recordings. The software in the Atos Professional environment allows users to record a script.

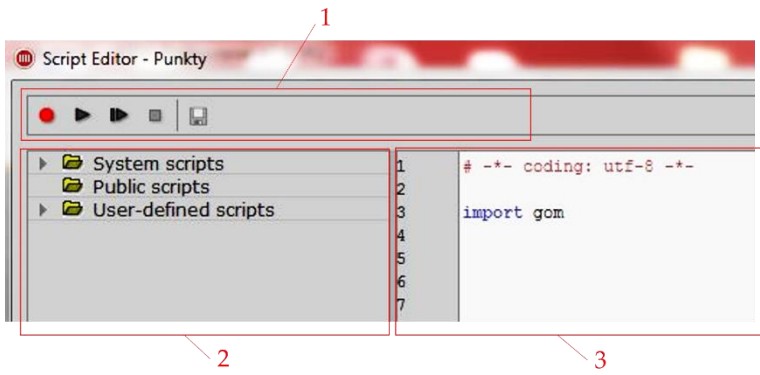

**Figure 8.** Creating scripts in Atos Professional.

In the top bar (Figure 8, reference 1) the editor contains buttons for recording scripts: record, start and stop. On the left side (Figure 8, reference 2) there is a catalog of preinstalled and user-defined scripts. There you can find items for creating new scripts, for renaming and removing items, just as you would with regular documents. It is also possible to export and import scripts and script archives. The scripts directory consists of three parts. The first contains preinstalled system scripts, the second is a public folder for multi-user access, and the third contains the user's private scripts. The editor is used to modify the currently selected script, in this section the block structure of the script is also displayed. In addition to the common operations (copy, paste, find and replace), the context menu of the editor also contains analog recorder buttons. The third part of the script editor (Figure 8, reference 3) is the exit area. The results obtained during the script execution are displayed there, see Figure 9.

```
import gom

MCAD_ELEMENT=gom.script.section.create_multisection_by_pa
rallel_planes (
        direction_mode='symmetric',
        name='Section 1',
        position=0.00000000e+00,
        properties=gom.Binary ('eAHtV11o...'),
        reference_plane=gom.app.system['system_plane_x'],
        section_distance=5.00000000e+01,
        separated_elements=False)
```

**Figure 9.** One of the recording results.

Initially, the function is selected from the gom library, which is divided into several layers. Therefore, a few additional parameters are needed to achieve the create_multisection_by_parallel_planes function. An open parenthesis at the end of a line indicates the start of a function call. The following lines are the parameters one per line. The closing brace indicates the end of the function call. Everything is assigned to the MCAD_ELEMENT variable.

The following data types are used as parameter values:

- Strings (direction_mode, name);
- Whole numbers (num_sections);
- Floating point numbers (position and section_distance);
- Boolean values (separated_elements).

For research purposes, a script was developed that implements the process of measuring, determining deviations and sending the obtained values. It is possible to change the script and the developer can adapt it to their needs.

For the correct implementation of grinding, the amount of allowance to be removed in the given areas must be defined. The 3D scan (point cloud) obtained as a result of the

measurement was compared with the reference CAD model and the surpluses at 80 points were determined using geometric operations in the places shown in Figure 10.

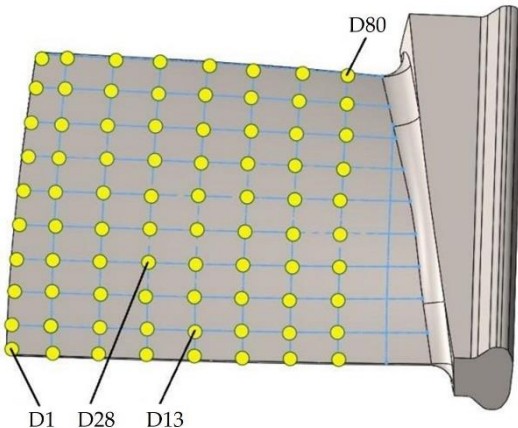

**Figure 10.** The back of the shoulder blade with a grid of points.

In this way, the procedures performed for 80 points D1, D2, ... D80, which were registered in the script, were prepared shown in Figure 11.

```
import gom
import os
import socket

#D1
MCAD_ELEMENT=gom.script.inspection.create_distance_by_2_points (

MCAD_ELEMENT=gom.script.inspection.inspect_dimension (
#D2
MCAD_ELEMENT=gom.script.inspection.create_distance_by_2_points (

MCAD_ELEMENT=gom.script.inspection.inspect_dimension (
#D3
MCAD_ELEMENT=gom.script.inspection.create_distance_by_2_points (

MCAD_ELEMENT=gom.script.inspection.inspect_dimension (
                                    .
                                    .
                                    .
#D80
MCAD_ELEMENT=gom.script.inspection.create_distance_by_2_points (

MCAD_ELEMENT=gom.script.inspection.inspect_dimension (
```

**Figure 11.** Script with recorded allowances.

The recorded script starts with the lines: gom, os, socket, which are responsible for including the Python libraries. The gom library is responsible for cooperation with Atos Professional software and the user does not have access to it, but can be used in the recording option. The os library refers to the options of the currently used operating system. Next, the script is the results of activities on the point cloud obtained from measurements with a 3D scanner. After saving the allowances, the data are sent from the Atos Professional client to the Matlab server in order to select the pressure forces with a neural decision-making system [32].

## 5. The Selection of the Pressure Force with a Neural Decision-Making System

The purpose of the developed system is to select the force in contact of the blade with the electro-spindle tool during grinding. The pressure is carried out by an industrial robot using the Force Control add-on. The grinding pressure force is generally presented as:

$$F_n = f(Q, x_C, y_C, z_C) \tag{1}$$

In order to avoid the problem of modeling the relationship between the force and the allowance $Q$ at the points $x_C$, $y_C$, $z_C$ on the blade, an artificial neural network was used, which was taught the relationship (1) with the following assumptions:

- constant rotational speed of the electro-spindle tool $n_t$ = 4500 rpm;
- constant feed speed (blade speed relative to the tool) $v_r$ = 0.02 m/s;
- variable pressure force of the blade against the tool in the range of 2–10 N.

After the assumptions, two variables were obtained which determine the pressure force of the blade against the tool:

$$F_n = f(Q, \ n) \tag{2}$$

where $Q$—allowance, $n$—number of a point on the D1, D2, . . . D80 blade instead of the $x_C$, $y_C$, $z_C$ coordinates.

Using Matlab software and "Neural Network Toolbox" libraries, a neural controller of the blade grinding process was realized. In order to approximate the tool pressure force from the material allowance, a feed-forward neural network was used, learned in accordance with the error back propagation algorithm [33]. The research was conducted and based on them, a network with two hidden layers with nine neurons in each of these layers was selected. In the hidden layers, sigmoid unipolar neuron activation functions were used, and in the output layer, linear activation functions were used. The weights of the neural network were learned according to an algorithm based on the Levenberg–Marquardt optimization method.

The data for training the artificial neural network were obtained during measurement experiments. In 80 points located on the surface of the ridge of the blade, geometrical measurements were made in order to determine the size of the material allowance shown in Figure 7. On this basis, the dependence of the thickness of the collected material layer on the tool pressure force was determined. During the tests, the applied values of the pressure forces were: 2, 3, 4, 5, 6, 7, 8, 10 N. The discussed tests were carried out randomly on 30 blades in the manner presented in Table 1.

**Table 1.** The structure of learning neural networks.

| | | Pressure [N] | | | | | | | | | | | |
|---|---|---|---|---|---|---|---|---|---|---|---|---|---|
| | | 2 | | | | 3 | | | . . . | 10 | | | |
| **Point no.** | 1 | Thickness of collected allowances | | | | Thickness of collected allowances | | | . . . | Thickness of collected allowances | | | |
| | 2 | | | | | | | | | | | | |
| | . . . | | | | | | | | | | | | |
| | 80 | | | | | | | | | | | | |
| | | Blade 1 | Blade 2 | . . . | Blade 30 | Blade 1 | Blade 2 | . . . | Blade 30 | . . . | Blade 1 | Blade 2 | . . . | Blade 30 |

The task of the neural process controller is to generate at the selected D1, D2, . . . D80 measuring points on the blade (Figure 10) the values of the pressure forces against the tool, if the value of the allowance at these points is known. The structure of the input and output layers of the neural network results from the structure of the measurement data. The neural

network structure has 80 inputs and 80 outputs as in Figure 12, which is as many as there are measurement points and a visible threshold value—b.

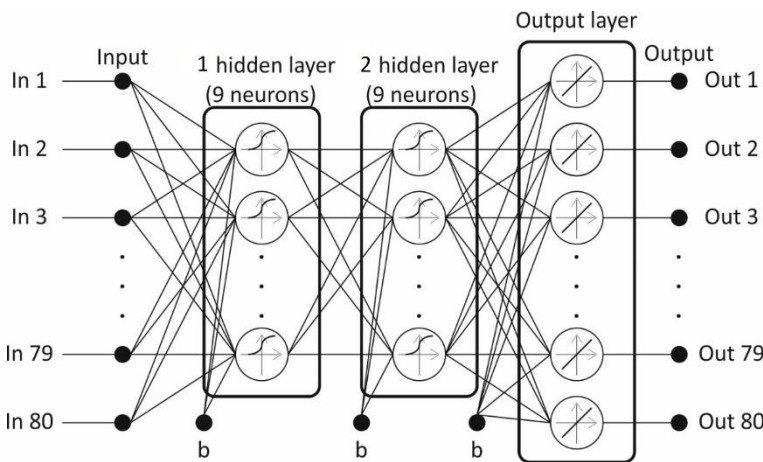

**Figure 12.** The structure of a neural network.

The system is built on the basis of an artificial neural network, which trains the relationship between the pressure force and the material allowance to be removed at individual points of the blade D1, D2, ... D80. After the neural decision-making system determines 80 values of the forces for individual points, they are sent to the robot's IRC5 controller in accordance with the communication parameters as in Figure 13.

```
#Communication parametrs
HOST = '192.168.1.105' #server adress
PORT = 9000
BUFFER_SIZE = 1024

#Saving distance D1 as variable
dyst1 = gom.app.project.inspection[ 'D1' ].get ( 'scalar_value' )
dyst_string = "{ }" .format (dyst1)
print (dyst_string)

#TCP/IP communication
DATA1 = dyst_string.encode ( 'UTF-8' ) #Data prepared for sending
s = socket.socket(socket.AF_INET, socket.SOCK_STREAM)
s.connect ( ( HOST, PORT) )
print ( "Connected. . . " )
s.send (DATA1)
data = s.recv (BUFFER_SIZE) #Received data
print ( "Received: { } " . format (data.decode ( 'UTF-8' ) ) )
s.close ( )
print ( "Connection closed. " )
```

**Figure 13.** Definition of communication parameters.

Communication parameters must be defined: server address, port and buffer needed to receive data. Then, the determined force values are taken and saved to a variable. It has to be converted to byte format because it is sent over the TCP/IP protocol. After establishing communication with the robot controller, the data determined by the neural network are sent. After the data have been transferred, the connection is closed. The transferred data are processed and the automatic change in pressure force of the workpiece to the grinding wheel is achieved.

## 6. Automatic Correction of the Pressure Force for the Programmed Path

The robotic process of grinding or polishing is performed with the use of belt grinders [4,15,34,35] or rotary tools [36,37]. A novelty is the development of a robotic

grinding process that will take into account the individual geometrical parameters of the detail (e.g., blades) on the basis of automated measurements obtained with a 3D optical scanner and determination of processing parameters with a neural decision system. A solution was chosen in which the robot with the force control option holds the blade and presses it against the tool. A rotary grinding tool was used as the machining tool. The non-contact measurement of the blade geometry is performed with the use of a 3D scanner. To control the process, a superior system generating the set pressure force of the blade against the tool was used.

Figure 14 shows graphically the values of the pressure forces generated by the neural process controller at individual measuring points, which in the next process become points belonging to the programmed robot path.

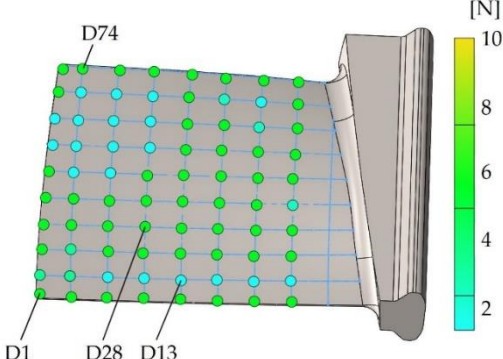

**Figure 14.** Graphical presentation of the pressure forces at individual points on the blade.

Depending on the allowance at the measuring points on the blade, a pressure force was generated. In the case of blade processing, adaptation of the robot's trajectory to the changing shape is required. This can be ensured by one of ABB's additions, namely the Force Control package and the FC Pressure option used by the authors. After the tests, a solution was proposed in which a stationary tool was used (electrospindle with a grinding wheel), and the object reference system (WorkObject) associated with the robot arm (blade in the gripper) was movable. The advantage of this solution is the reduction in disturbance level, because when the electrospindle is located on the robot's arm, the force value is recorded with disturbances resulting from dynamic phenomena.

A view of the robot application with a blade, set motion trajectory and directions of forces in RobotStudio are presented in Figure 15. The use of the approach with force control allows for the adaptation of the trajectory (change of the pressure force at selected points) to the shape of the part, which varies to a certain extent.

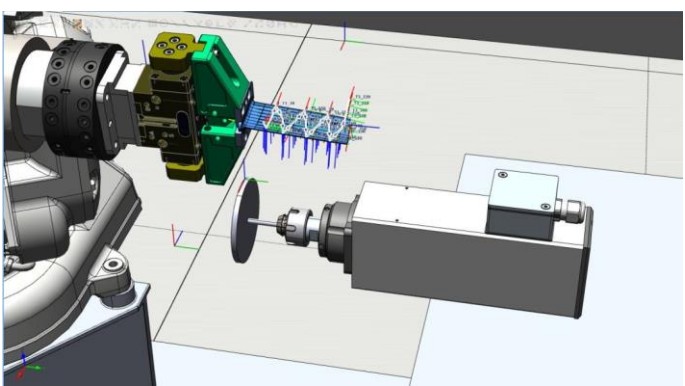

**Figure 15.** Application view in RobotStudio.

### 7. Machining

Tests were conducted on the designed robotic station with a 3D measuring system, a neural process controller and a robot with force control. The developed algorithm of the robotic machining process is shown in Figure 16.

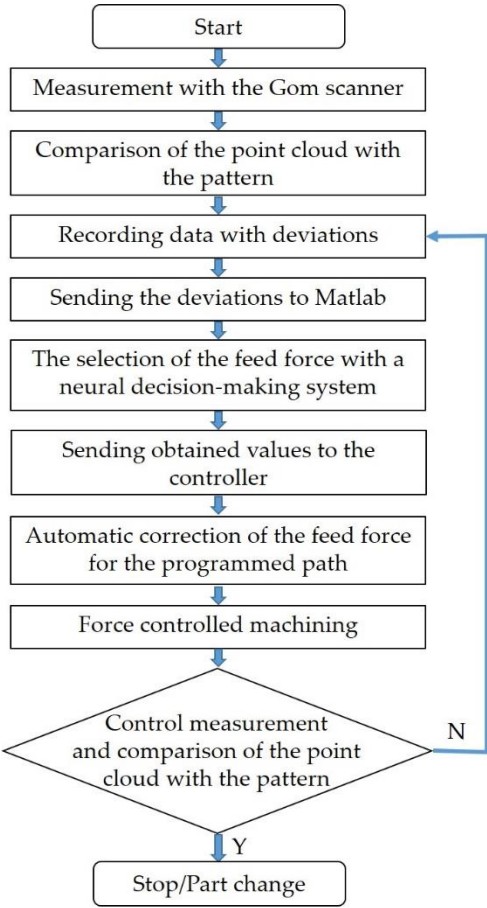

**Figure 16.** Information flow algorithm in robotic processing in correlation with a 3D scanner and a neural decision-making system.

The process begins by measuring the blade geometry and comparing it with a standard in Atos Professional to determine the deviation grid at 80 points. The saved deviations are sent to the Matlab software for the selection of the pressure forces by the neural decision system for each point. The obtained pressure force values are sent to the robot controller and automatically adapted to the programmed path at selected points. In the next stage, the machining process begins (robot grinding with the option of force control) along the programmed path. After the grinding process, the geometry is measured again and the allowances are determined. When the measured allowances are within the tolerated limits, the grinding is completed and the blade is deposited and the next one is taken. When the measured allowances are outside the tolerated limits, the blade grinding process is repeated until the assumed dimensions are achieved. The entire process of processing the blades was tested on the designed station and the visual results were obtained as in Figure 17, and the measurement of the allowance as in Figure 18.

When comparing the CAD model of the blade with the measurements obtained with a 3D scanner, the tolerance at the measurement points should not exceed $\pm\,0.05$ mm. The conducted verification tests confirm the correctness of the adopted objectives regarding measurements, determining the value of pressure forces from the measured allowances and machining in a robotic process.

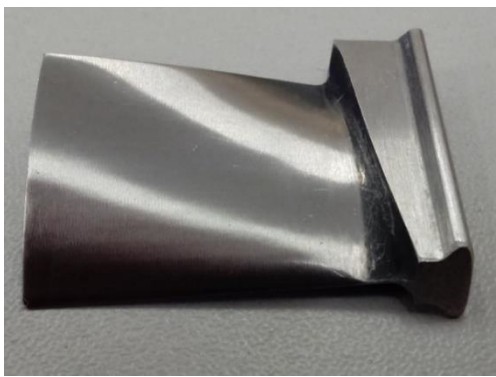

**Figure 17.** View of the blade after grinding.

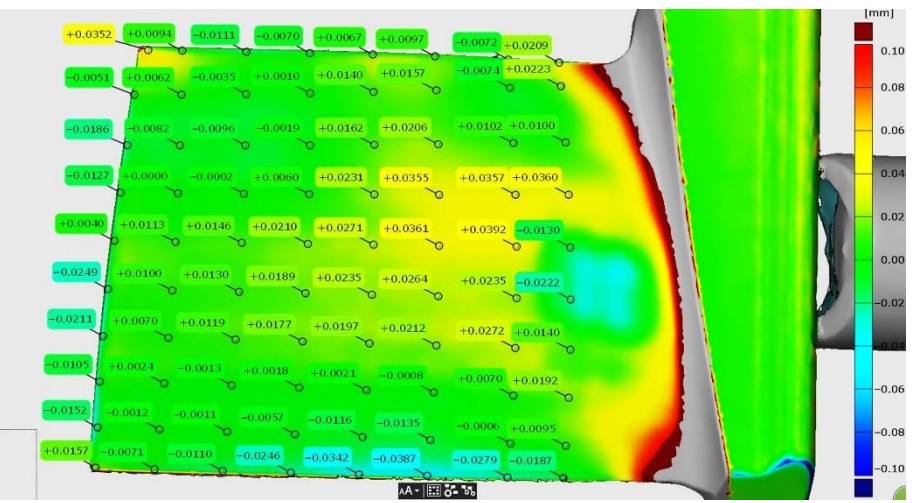

**Figure 18.** The measured allowances after grinding.

Figure 19 shows the results of the measurement after machining for the blade, namely its thickness in cross-sections (tolerance $\pm$ 0.2 mm) for the control allowing the blade to be installed in the engine.

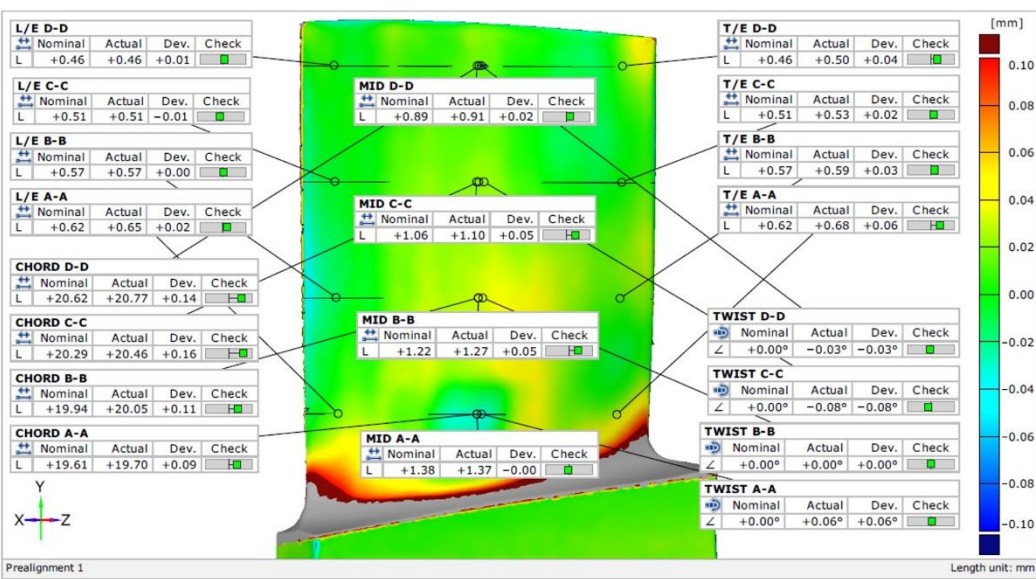

**Figure 19.** Results of the measurement of the blade in control sections.

Other aviation components subjected to the developed process include deburring the edges of the ADT housing with measurement of the radius repeatability shown in Figure 20, and deburring on the bosses with the measurement of the obtained chamfer shown in Figure 21.

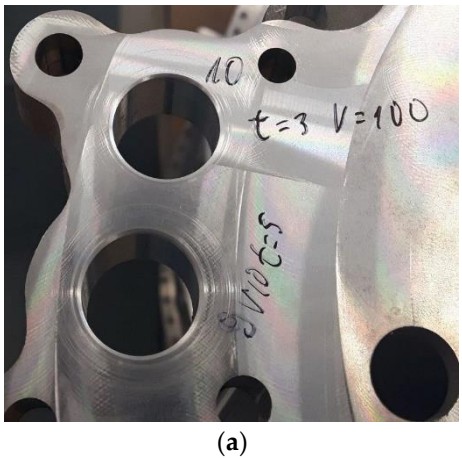

(**a**)

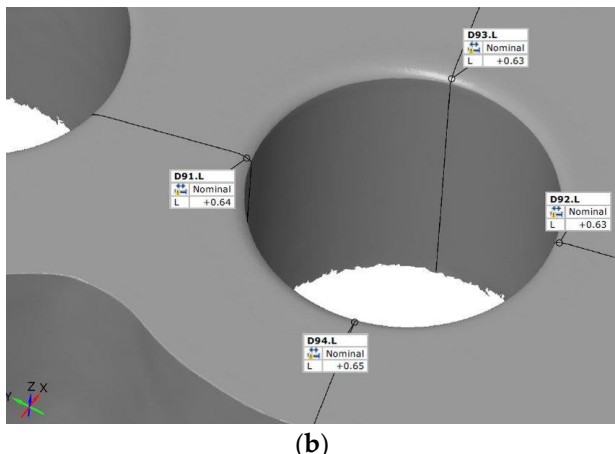

(**b**)

**Figure 20.** Deburred ADT gearbox housing: (**a**) block section; (**b**) measuring the repeatability of the radius.

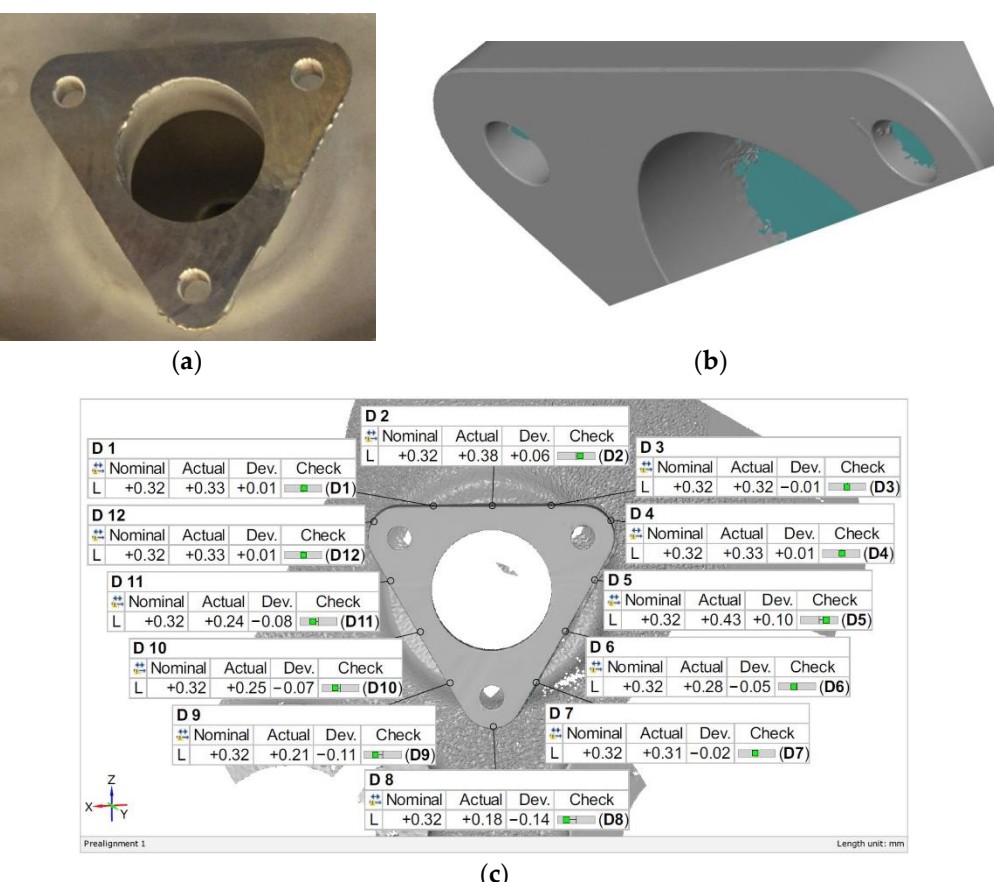

(**c**)

**Figure 21.** Deburred boss: (**a**) before the process; (**b**) after the process; (**c**) measurement of the obtained chamfer.

The presented approach allows for adaptive machining by determining the pressure force of the blade against the tool (grinding wheel) depending on the measured allowance. This avoids a situation in which the pressure force is constant over the entire surface of the

blade, which can cause a large allowance in some areas of the blade or damage it in the case of large losses.

## 8. Conclusions

The paper presents the proprietary application of a 3D scanner and a method of communication containing a script sent via TCP/IP protocol in robotic processes of aviation components. The innovative approach is based on conducting a measurement, saving the results, sending their size (allowances), comparing them with the pattern to the decision system and then to a robot controller for adaptive processing by a robot with force control. The proposed solution was verified on three objects: in the selection of parameters during blade grinding, in the process of deburring bosses and the edges in the ADT housing. A set of experimental data was employed in the research, by means of which the relationship between the location of a point on the trajectory of the blade, the pressure force and the thickness of the collected material layer was determined. An artificial neural network was used to determine the variable pressure force of the blade by the robot against the tool (grinding wheel) on the basis of the measured allowances. The presented solution was implemented for the needs of and in cooperation with Pratt & Whitney Rzeszów S.A.

**Author Contributions:** Conceptualization, D.S. P.G. and K.K.; methodology, K.K. P.G. and D.S.; software, A.B. P.G. and K.K.; validation, M.M. and A.O.; formal analysis, M.M. and M.U.; investigation, A.B.; resources, K.K.; data curation, M.U.; writing—original draft preparation, P.G.; writing—review and editing, M.M. and K.K.; visualization, P.G. and K.K.; supervision, A.B.; project administration, P.G.; funding acquisition, A.B. All authors have read and agreed to the published version of the manuscript.

**Funding:** This research received no external funding.

**Institutional Review Board Statement:** Not applicable.

**Informed Consent Statement:** Not applicable.

**Data Availability Statement:** Data are contained within the article.

**Conflicts of Interest:** The authors declare no conflict of interest.

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
