# Peer review of "Application of a 3D Scanner in Robotic Measurement of Aviation Components"

_electronics, doi:10.3390/electronics11193216_

Round 1
Reviewer 1 Report
- In the introduction section, it was mentioned the previous works and did not focus on the motivation of the presented work. Add the motivation of the presented manuscript with some details,
Reviewer 2 Report
1.what is the accuracy requirement of the 3D point?Is its application scene constrained by the environment and lighting?
2.Could the author please explain using artificial neural network to calculate the grinding pressure? Is there another way?
3.Could the author please explain the advantages of pressure feedback grinding over non-pressure feedback grinding?
4.Could the pressure be changed continuously and the sample surface smoother during grinding if it is set at 2, 4, 6, 8, 10N?
5.Could the author please explain the advantages of TCP/IP over other protocols?
6. There is plenty of room to explain how to use Atos software to obtain point cloud data. It is suggested that the author please shorten the length of this section.
Reviewer 3 Report
The structure and the descriptions of this paper are well. However, there are still some suggestions for the authors.
1. For Figure 1, the vision plane is marked but the marked height is not from the vision plane. This is different from what we usually think of graphical descriptions. If this is correct, it should be further explained.
2. Some of the writing in the text does not meet the grammatical requirements. For example, regarding the description of a figure (say Figure 1), "Figure X" is inserted into the description without any conjunctions, resulting in a violation of English grammar.
